# Glycoprotein N-linked glycans play a critical role in arenavirus pathogenicity

Takaaki Koma[1¤], Cheng Huang[1], Adrian Coscia[2], Steven Hallam[1], John T. Manning[1], Junki Maruyama[1], Aida G. Walker[1], Milagros Miller[1‡], Jeanon N. Smith[1], Michael Patterson[1], Jonathan Abraham[2], Slobodan Paessler[1]*

**1** Department of Pathology, University of Texas Medical Branch at Galveston, Texas, United States of America, **2** Department of Microbiology, Harvard Medical School, Boston, Massachusetts, United States of America

¤ Current address: Department of Microbiology, Tokushima University Graduate School of Medical Science, Tokushima, Japan
‡ Unavailable to confirm authorship.
* slpaessl@utmb.edu

**Data Availability Statement:** All relevant data are within the manuscript and its Supporting Information files.

## Abstract

Several arenaviruses cause hemorrhagic fevers in humans with high case fatality rates. A vaccine named Candid#1 is available only against Junin virus (JUNV) in Argentina. Specific N-linked glycans on the arenavirus surface glycoprotein (GP) mask important epitopes and help the virus evade antibody responses. However the role of GPC glycans in arenavirus pathogenicity is largely unclear. In a lethal animal model of hemorrhagic fever-causing Machupo virus (MACV) infection, we found that a chimeric MACV with the ectodomain of GPC from Candid#1 vaccine was partially attenuated. Interestingly, mutations resulting in acquisition of N-linked glycans at GPC N83 and N166 frequently occurred in late stages of the infection. These glycosylation sites are conserved in the GPC of wild-type MACV, indicating that this is a phenotypic reversion for the chimeric MACV to gain those glycans crucial for infection *in vivo*. Further studies indicated that the GPC mutant viruses with additional glycans became more resistant to neutralizing antibodies and more virulent in animals. On the other hand, disruption of these glycosylation sites on wild-type MACV GPC rendered the virus substantially attenuated *in vivo* and also more susceptible to antibody neutralization, while loss of these glycans did not affect virus growth in cultured cells. We also found that MACV lacking specific GPC glycans elicited higher levels of neutralizing antibodies against wild-type MACV. Our findings revealed the critical role of specific glycans on GPC in arenavirus pathogenicity and have important implications for rational design of vaccines against this group of hemorrhagic fever-causing viruses.

## Author summary

Several arenaviruses cause severe hemorrhagic fevers in humans. The only vaccine against arenavirus infections is Candid#1, a live attenuated vaccine against Argentine hemorrhagic fever. So far, we have successfully attenuated additional one of the arenaviruses,

**Funding:** T.K. and J.M. were supported by in part by a JSPS Postdoctoral Fellowship for Research Abroad (No. H28-803 and No. H29-296, respectively). J.A. is a recipient of a Burroughs Wellcome Fund Career Award for Medical Scientists (2016). The funders had no role in study design, data collection and analysis, decision to publish, or preparation of the manuscript.

**Competing interests:** The authors have declared that no competing interests exist. Author Milagros Miller was unable to confirm their authorship contributions. On their behalf, the corresponding author has reported their contributions to the best of their knowledge.

Machupo virus, the causative agent of Bolivian hemorrhagic fever. Unraveling this attenuation mechanism might help the development of live-attenuated vaccines for other arenaviruses. In this study, we revealed that the specific glycans of the viral glycoproteins play an important role in pathogenicity *in vivo*. The glycans facilitate the virus to evade neutralizing antibodies. This study would contribute to the development of arenavirus vaccine candidates.

## Introduction

Several arenaviruses cause hemorrhagic fevers in humans and are serious public health concerns. These include Machupo (MACV), Junin (JUNV), Guanarito (GOTV), Sabia (SABV), Chapare, and Lassa (LASV) viruses [1,2]. MACV is a New World arenavirus and causes Bolivian hemorrhagic fever (BHF), a zoonotic disease that is endemic in Bolivia [3–5]. MACV often persistently infects its natural host, *Calomys callosus* [6]. Infection in humans has a case fatality rate of 25% to 35% [3,4,7]. Initial symptoms include fever, malaise, myalgia, headache, and anorexia. During the second week of illness, approximately one third of cases develop severe neurological and/or hemorrhagic symptoms. BHF emerges and re-emerges only in the endemic area [4]. Several arenaviruses, including MACV, are classified as select agents by the U.S. Department of Health and Human Services. There are currently no FDA-approved vaccines or drugs for BHF [8,9]. Thus, there is an urgent need for a vaccine against MACV to protect individuals that are at a high risk of infection and as a countermeasure against potential bioterrorism.

MACV belongs to the *Arenaviridae* family and contains a bi-segmented (L and S segments) ambisense RNA genome [10]. The L segment encodes the RNA dependent RNA polymerase (L) and the RING finger protein (Z). The S segment encodes the viral glycoprotein precursor (GPC) and the nucleoprotein (NP). Cellular signal peptidase cleaves GPC into a stable signal peptide (SSP) and GP1/GP2, and cellular subtilase SKI-1/S1P further cleaves GP1/GP2 into the GP1 and GP2 subunits [11–15]. GP1, the N-terminal subunit, mediates cellular receptor binding and is also the target of host neutralizing antibodies [16]. The GP1 subunits of the pathogenic New World arenaviruses MACV, JUNV, GOTV and SABV, bind human transferrin receptor 1 (hTfR1) [17–19]. The GP2 subunit at the C-terminal contains a transmembrane domain (TMD) and cytoplasmic tail (CT), and mediates fusion of viral and host cell membranes after particles are internalized into acidified endosomes [12]. The ectodomain of arenavirus GPCs contains several N-linked glycosylation motifs that are important for the expression, proper folding and cleavage of GPC [20–23]. For Old World lymphocytic choriomeningitis virus (LCMV), the addition or disruption of certain glycosylation sites on GPC is also shown to affect virus cell tropism and fitness in cultured cells [24]. Furthermore, it has been reported that specific glycans on arenavirus GPCs mask important epitopes and thereby facilitate viral evasion of antiviral activities of neutralizing antibodies [25–27]. However, the importance of GPC N-glycans in arenavirus pathogenicity has not been revealed. In this study, we report for the first time the critical role of specific N-glycans on GPC in the virulence of a hemorrhagic fever-causing arenavirus *in vivo*.

MACV is closely related to JUNV, the etiological agent of Argentine hemorrhagic fever (AHF) [28]. An effective vaccine against JUNV, the Candid#1 (Cd#1) live attenuated vaccine, is available in Argentina and has substantially reduced AHF cases within endemic areas [28,29]. The Cd#1 strain was developed by passaging the pathogenic JUNV XJ strain twice in guinea pigs and 44 times in suckling mice brains [30,31]. The resulting strain (XJ44) was

attenuated in guinea pigs but still virulent in suckling mice. Between passages 13 to 44, a T168A substitution occurred on GP1 that led to the loss of an N-linked glycan at GP1 N166 [32,33]. XJ44 was further passaged in cultured cells, where a Phe to Ile mutation at residue 427 (F427I) occurred in the TMD of JUNV GPC. The resulting Cd#1 strain is attenuated in suckling mice. The F427I substitution alone is important but not sufficient for full attenuation of JUNV; when the F427I change was introduced into the GPC of pathogenic JUNV, the mutant virus (JUNV GPC$_{F427I}$) was attenuated in suckling mice and guinea pigs but still resulted in 10% lethality in suckling mice and caused mild disease manifestations in guinea pigs. Likewise, MACV GPC$_{F438I}$ (equivalent to JUNV GPC$_{F427I}$) was only partly attenuated in a mouse model of MACV infection [34]. These observations suggested that multiple changes in GPC are required for optimal attenuation.

We previously found that recombinant JUNV or MACV expressing the entire Cd#1 GPC in place of their respective wild-type GPCs was highly attenuated and immunogenic *in vivo* [35,36]. To identify additional virulence determinants, we then exchanged the MACV GPC ectodomain with the Cd#1 GPC ectodomain while retaining the MACV GP2 TMD and CT. The resulting recombinant virus, named MCg1, was partially attenuated in a IFN-αβ/γ R$^{-/-}$ mouse model of MACV infection, indicating the ectodomain of MACV GPC also contains virulence determinants [37]. In this study, we analyzed the sequence of the MCg1 viruses isolated from infected animals and frequently identified two substitutions (P85S and A168S/T) that would allow MCg1 to acquire N-linked glycans in GP1 at positions N83 and N166. The P85S and A168S/T mutations in MCg1 GP1 correspond to the N83-X-S85 and N166-X-T168 glycosylation motifs in MACV GP1, respectively. To determine the significance of these substitutions, we introduced the P85S and A168S/T mutations, either alone or in combination, into MCg1. Acquisition of these glycosylation sites allowed the MCg1 virus to gain virulence in animals to an extent comparable to that of wild-type virus. Furthermore, we found that disruption of the specific glycosylation sites on MACV GPC led to attenuation of the highly pathogenic arenavirus in animals without noticeable influence on the expression and cleavage of GPC or viral fitness in cultured cells. Those mutant viruses were more sensitive to antibody neutralization presumably due to loss of the glycans that mask important epitopes involved in virus entry. Interestingly, infection of animals with MACV mutants lacking specific glycans on GPC elicited higher levels of neutralizing antibodies to wild-type MACV, suggesting these glycans have an impact on GPC immunogenicity. These new findings may inform the development of vaccine candidates against pathogenic arenavirus infections.

## Results

### Mutations enabling MCg1 virus to acquire N-linked glycosylation sites on GP1 emerge in infected animals

In our previous experiments, MCg1 (which has the ectodomain of Cd#1 GPC in a MACV backbone, Fig 1A) was partially attenuated in IFN-αβ/γ R$^{-/-}$ mice. MACV and Cd#1 GPCs have 9 and 7 N-linked glycosylation sites, respectively (Fig 1A). Among these glycosylation sites, the N83, N137, and N166 sites are unique to MACV GPC. As with MACV, pathogenic JUNV strains (Romero and XJ13) also contain the N166 glycosylation site in GPC. The N166 glycan, however, was lost in Cd#1 GPC during serial passage for vaccine development. We sequenced virus isolated from animals that succumbed to infection and from surviving animals at a later stage. Our sequencing analysis identified the C341T and/or G590A/T nucleotide substitutions in the MCg1 S genome (Fig 1B) in 6 out of 7 infected animals (S1 Table). These mutations resulted in the P85S and the A168S/T amino acid sequence changes in MCg1 GP1. Both mutations introduced a sequon (N-X-S/T) motif that would potentially result in glycans

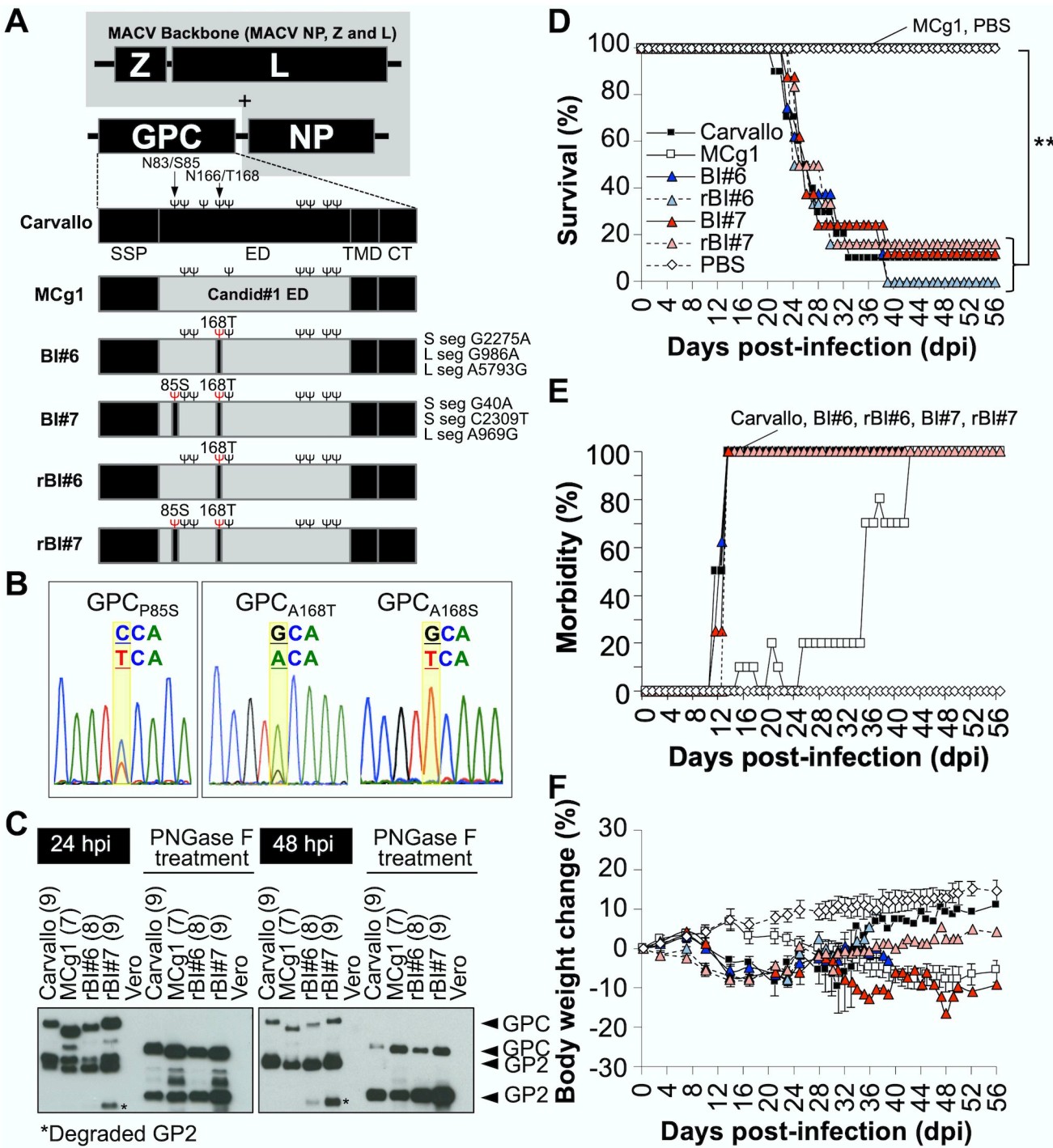

**Fig 1. MACV Carvallo strain expressing Cd#1-GPC ectodomain with reversions.** (A) Schematic representation of the genome of MCg1 and its revertants. BI#6 and BI#7 have synonymous substitutions as described on the right side of the GPC schematic. Ψ in black character represents an N-glycosylation site. Ψ in red character indicates an N-glycosylation site obtained by reversion. SSP: stable signal peptide, ED: ectodomain, TMD: transmembrane domain, CT: cytoplasmic tail. (B) Representative sequence chromatograms at reverted sites on GPC of MCg1 recovered from brain of MCg1-infected mice. (C) The presence of N-glycosylations on GPC was validated by WB. Vero cells were infected with each virus at MOI = 1. Cell lysates were treated with PNGaseF to remove N-linked glycosylations. Number of putative N-glycosylation sites was indicated in parentheses. (D) Survival rate of the IFN-αβ/γ R$^{-/-}$ mice after infections (Carvallo N = 10, MCg1 N = 10, BI#6 N = 8, rBI#6 N = 6, BI#7 N = 8, rBI#7 N = 6, PBS N = 6). Statistically significant differences are indicated by asterisks (**, $P<0.01$ by log rank test). (E) Morbidity rate of the IFN-αβ/γ R$^{-/-}$ mice after infections. Scruffy coats, hunched posture, lethargy, imbalance, partial paralysis, not active and death were counted as a sign of disease. (F) Body weight changes were monitored on the indicated days. Error bars indicate the SEM. The data shown are pooled from two independent experiments except rBI#6 and rBI#7.

being added to N83 and N166 on GPC. The A168S/T mutation in MCg1 GP1, which should lead to glycosylation of N166, specifically reversed the T168A substitution in Cd#1 GP1 to the sequence of its parental XJ13 strain, which contains this glycan.

Our sequencing data also suggested a heterologous virus population in the same hosts, as we observed double peaks at the same positions in chromatograms (Fig 1B). We used TA cloning of PCR fragments and sequencing to determine the frequency of nucleotide changes at each position (nt 341 and nt 590). We did not detect these mutations in samples collected at 17 days post infection (dpi). At later time points, the parental MCg1 virus could be found in the tissue homogenates in only 3 of 7 animals (#1, #2 and #4) (S1 Table). Two of the animals (#2 and #4) had virus populations with P85S and A168S/T substitutions in GPC. 5 of 7 animals (#2, #4, #5, #6 and #7) contained populations either with single amino acid change at A168 (GPC$_{A168S/T}$) or with double changes at P85 and A168 (GPC$_{P85S}$ and GPC$_{A168S/T}$). Animals #5 and #7 had viruses not only with Thr but also a Ser substitution at position 168 in tissue homogenate (Fig 1B), both of which could lead to acquisition of an N-linked glycan at N166. These results indicated that there is a strong selective pressure for gaining N-linked glycans at N83 and N166 *in vivo*.

## Acquisition of GPC N-glycan at N83 and N166 enhances the virulence of MCg1 *in vivo*

To study the virulence of MCg1 mutants in the MACV mouse model, we isolated MCg1 from brains of infected animals after two successive plaque purifications. We recovered two brain isolates; one with the GPC$_{A168T}$ substitution (BI#6) and the other with GPC$_{P85S}$ and GPC$_{A168T}$ substitutions (BI#7). MCg1 with GPC$_{P85S}$ could not be isolated. As shown in Fig 1A, the BI#6 and BI#7 isolates also contained three silent mutations in other regions of the viral genome. To determine the impact of acquisition of glycans on virulence, we used reverse genetics to rescue recombinant (r) rBI#6 and rBI#7 without the three silent mutations. We tested glycan site occupancy using Western blotting (WB) staining for GPC precursor and GP2 in infected cells (Fig 1C). The GPC rBI#6, which could be glycosylated on N83, migrated more slowly than the MCg1 GPC, suggestive of a higher molecular weight species. rBI#7 GPC, which could be glycosylated on N83 and N166, also migrated more slowly than MCg1 and rBI#6 GPC. Compared to the GP2 of MCg1, rBI#6 and rBI#7 migrated similarly, indicating that the migration difference for GPCs was due to the GP1 region. After treatment with PNGase F, a glycosidase that removes N-linked glycans, the migration of deglycosylated MCg1, rBI#6, and rBI#7 GPCs was similar. Taken together, these data demonstrated that the GPCs of rBI#6 and rBI#7 acquired additional glycosylation sites at N166 and N83/N166, respectively, as the result of A168T and P85S/A168T substitutions.

We next tested the virulence of the MCg1 mutants in animals. IFN-αβ/γ R$^{-/-}$ mice were intraperitoneally inoculated with the rBI#6 and rBI#7 isolates and monitored for survival, disease signs and virus dissemination for 56 days. As expected, MCg1-infected animals did not succumb to infection. In contrast, most of the BI#6-, rBI#6-, BI#7- and rBI#7-infected animals succumbed to the infection or reached a humane endpoint at 21–39 dpi, similar to wild-type infected animals (Carvallo strain), which were used as a positive control (Fig 1D). These data indicate that the acquisition of an N-glycan at GPC166 enhanced the virulence of the MCg1 virus in animals.

Most of the Carvallo-, BI#6-, rBI#6-, BI#7- and rBI#7-infected animals started to lose more than 5% of their body weight at 10 to 14 dpi and developed other disease signs, including scruffy coats and hunched posture at 11 to 20 dpi (Figs 1E and 1F and S1). Neurological signs such as imbalance were observed in these animals prior to death, similar to those of some

patients with BHF [38]. In MCg1-infected mice, a 7 to 34-day delay in 5% body weight loss was observed. Unlike Carvallo-, BI#6-, rBI#6-, BI#7- and rBI#7-infected animals, MCg1-infected animals showed only mild disease signs, without noticeable neurological signs until 46 dpi (S1 Fig). However, 60% of the MCg1-infected animals developed severe neurological disease signs such as imbalance and partial paralysis after 46 dpi. This might be due to the longer survival times of MCg1-infected mice, allowing more time for the virus to disseminate to the central nervous system. One animal from each of the Carvallo, BI#7 and rBI#7 group survived up to the the end of the study.

We examined virus dissemination in animals by measuring viral titers in brain, lung, and liver tissues and in serum (Fig 2A). Overall, viral titers were highest in the brains and lungs of infected animals. Among different infection groups, the Carvallo group had the highest viral titers in all organs tested, while the MCg1 group exhibited the lowest viral titers. The virus titers in BI#6-, rBI#6-, BI#7- and rBI#7-infected animals were comparable to that of the Carvallo group in brain and lung, and lower in liver and serum. The virus load in MCg1-infected animals was below the detection limit in liver and serum. We detected viral RNA in the brains of all infected animals, including animals in the MCg1 group. There were no significant differences in viral titers between the BI#6/rBI#6 groups and the BI#7/rBI#7 groups, but overall the titers in organs were slightly higher in BI#7 and rBI#7 groups than BI#6 and rBI#6 groups. The three silent mutations in BI#6 and BI#7 did not seem to impact the virulence in mice or viral titers compared with rBI#6 and rBI#7, respectively.

## The MCg1 with the $GPC_{P85S}$ and the $GPC_{A168S/T}$ substitutions readily emerged in infected animals

Because MCg1 mutants with the GPC P85S and the GPC A168S/T sequence changes appeared frequently in animals, we examined the occurrence of these changes in MCg1- and BI#6/rBI#6-infected animals by sequence analysis. We were able to detect viral RNA by RT-PCR in the brains of all infected animals, even though we were not able to detect infectious virus in some of the animals. 100% of the MCg1 isolates we sequenced had the $GPC_{A168S/T}$ substitution and 70% of isolates had the $GPC_{P85S}$ substitution (S2 Table). We did not, however, observe the $GPC_{P85S}$ substitution in BI#6 or rBI#6 (which contain the A168T change in GPC) infected animals, yet 70% of MCg1-infected animals had virus populations with the $GPC_{P85S}$ substitution.

## Gain of N-linked glycans on MCg1 GPC promoted antibody neutralization escape

To understand the potential impact of glycan acquisition on viral sensitivity to antibody neutralization, we performed $PRNT_{50}$ assays to compare the ability of serum from infected animals to neutralize MCg1, rBI#6, rBI#7 and MACV in vitro (Table 1 and Fig 3). MCg1 infection induced the strongest neutralizing antibody response in animals among these viruses. Serum samples from MCg1-infected animals had potent neutralizing activity against MCg1 and rBI#6, but relatively weaker activity against rBI#7 and Carvallo. Similarly, the sera from rBI#6-infected animals had potent neutralizing activity against MCg1 but weaker activity against rBI#6, rBI#7 and Carvallo. Sera from rBI#7-infected animals exhibited the strongest neutralizing activity against MCg1 but relatively weaker activity against the homologous virus or Carvallo. This indicates that the gain of N-linked glycans at N83 and N166 reduced MCg1's susceptibility to antibody neutralization. In another comparative experiment, we observed that MCg1 virus was more sensitive to neutralization by MCg1 sera than was rBI#6, and that rBI#6 was more sensitive than was rBI#7 (S2 Fig). Collectively, these data show that the number of

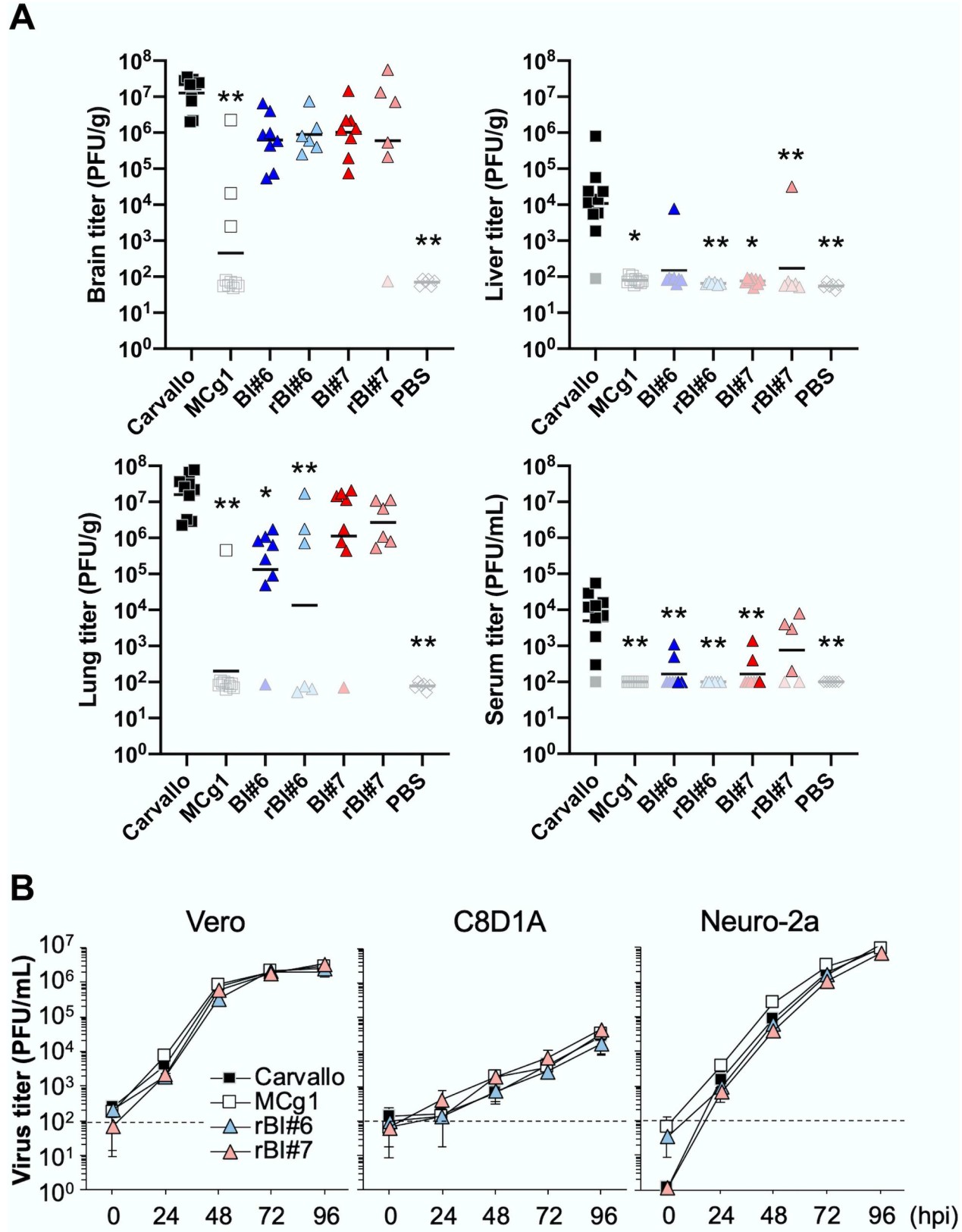

**Fig 2. Virus dissemination and the growth curves of the revertants.** (A) Virus titers in brain, lung, liver and serum. Each dot indicates an individual mouse. Samples under the detection limit were plotted with their detection limits in whitish color. The solid bar represents the geometric

mean of the titers. Statistically significant differences compared with Carvallo are indicated by asterisks (*, $P<0.05$ and **, $P<0.01$ by Dunn's post-test after Kruskal-Wallis test). (B) Virus growth of the revertants was characterized in African green monkey kidney-derived Vero cell, murine astrocyte-derived C8D1A cells and murine neuroblastoma-derived Neuro-2a cells (MOI = 0.01). The dashed line indicates the detection limit. Data shown are averages of triplicate wells with error bars indicating the SD.

glycans on GPC inversely correlates with virus sensitivity to antibody neutralization, which is consistent with previous results [26].

## N-linked glycans on GPC do not affect MCg1 virus growth in cell culture

To examine if the difference in virulence for MCg1 isolates could be attributed to virus growth *in vitro*, we characterized virus growth on African green monkey kidney (Vero) cells, murine astrocyte-derived C8D1A cells, and murine neuroblastoma-derived Neuro-2a cells. Although MCg1 grew slightly faster than other viruses at 24 and 48 hpi, all viruses reached similar titers at 72 and 96 hpi in all these cell lines (Fig 2B), indicating that the N-glycosylation on GPC has negligible impact on virus growth.

## Loss of GPC N-glycans at N83 and N166 attenuates MACV Carvallo

Since MCg1 gained virulence in mice after the acquisition of two N-linked glycosylation sites at N83 and N166, we reasoned that disruption of the corresponding N-glycosylation sites in pathogenic MACV Carvallo would attenuate the virus *in vivo*. To test this hypothesis, we introduced mutations in MACV GPC using a reverse genetics system to disrupt N-linked glycosylation sites either at N83 (MACV $GPC_{\Delta N83}$), N166 (MACV $GPC_{\Delta N166}$) or at both positions (MACV $GPC_{\Delta N83/N166}$) (Fig 4A). Our MCg1 infection data and previous studies show that the viruses often mutate to gain N-linked glycosylation sites (Fig 1)[24]. To minimize the likelihood of reversion, we simultaneously mutated the first (N) and the third (S or T) amino acid residues in the N-linked glycosylation motif (N-X-S/T). We confirmed the loss of N83 and N166 glycans on MACV GPC by WB analysis (Fig 4B). After that, we tested the susceptibility of these mutants to antibody neutralization using anti-Cd#1 sera obtained from C57BL/6J mice. Anti-Cd#1 sera did not effectively neutralize MACV Carvallo but strongly neutralized MACV $GPC_{\Delta N83/N166}$ and the homologous Cd#1 (Fig 4C). Our data clearly indicate that loss of N83 and N166 glycans on MACV GPC rendered the virus more susceptible to antibody

**Table 1.** PRNT$_{50}$ geometric mean titer.

| PRNT$_{50}$ geometric mean titer to | Serum group | | | | |
|---|---|---|---|---|---|
| | MCg1 | rBI#6 | rBI#7 | Carvallo | PBS |
| Carvallo virus | 104.5[a] (N = 10, 56 dpi)[b] | <30 (N = 6, 27.0 dpi) | 26.7 (N = 6, 30.8 dpi) | 25.7 (N = 9, 29.3 dpi) | <30 (N = 6, 56 dpi) |
| Carvallo virus (with serum collected at 56 dpi)[c] | 104.5 (N = 10, 56 dpi) | NA[e] | 120 (N = 1, 56 dpi) | <30 (N = 1, 56 dpi) | <30 (N = 6, 56 dpi) |
| MCg1 virus | 960< (N = 3[d], 33 dpi) | 960< (N = 6, 27 dpi) | 960< (N = 5, 31.4 dpi) | NA | NA |
| rBI#6 virus | 960< (N = 3, 33 dpi) | 604.8< (N = 6, 27 dpi) | NA | NA | NA |
| rBI#7 virus | 151.2 (N = 3, 33 dpi) | 106.9 (N = 6, 27 dpi) | 158.3 (N = 5, 31.4 dpi) | NA | NA |

a The titer <30 was calculated as 15 for geometric mean if samples contained <30. The titer was shown as <30 when all the samples were <30. The titer was shown as XXX.X< when samples contained 960<.

b (Numbers of samples tested, the average sample collection days after infection)

c Only the data with serum collected at 56dpi was displayed to align the collection date.

d To adjust the collection days after infection, the samples were picked up from the previous experiment.

e NA, Not available.

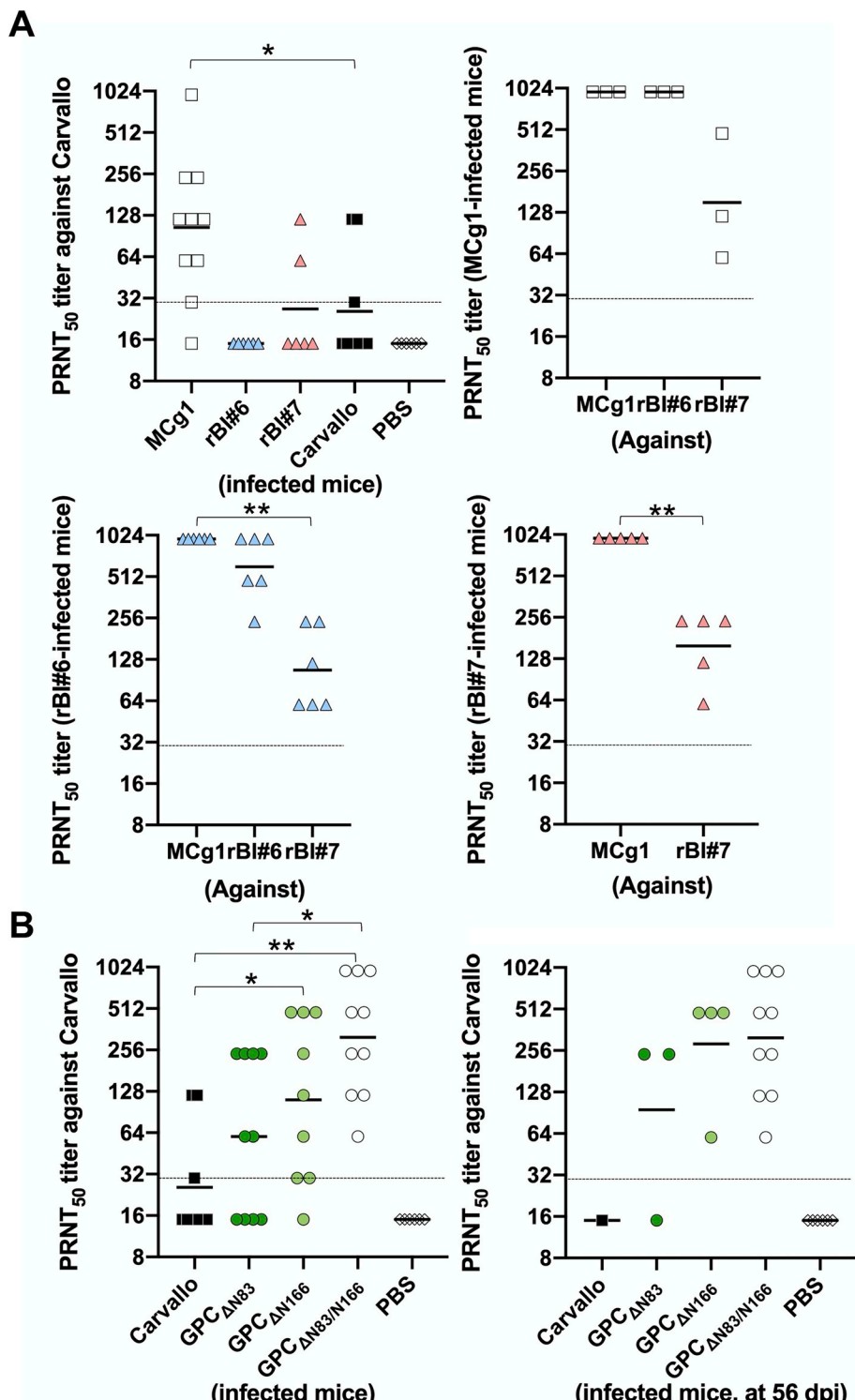

**Fig 3. Individual PRNT$_{50}$ titers.** (A) Individual PRNT$_{50}$ titers from Table 1. (B) Individual PRNT$_{50}$ titers from Table 2. The solid bar and the dashed line indicate the geometric mean of the titers and the detection limits, respectively. The serum dilution factor is 30 to 960 times, and PRNT$_{50}$ titer lower than 30 was plotted as 15. Statistically significant differences compared with Carvallo are indicated by asterisks (*, P<0.05 and **, P<0.01 by Mann-Whitney U test).

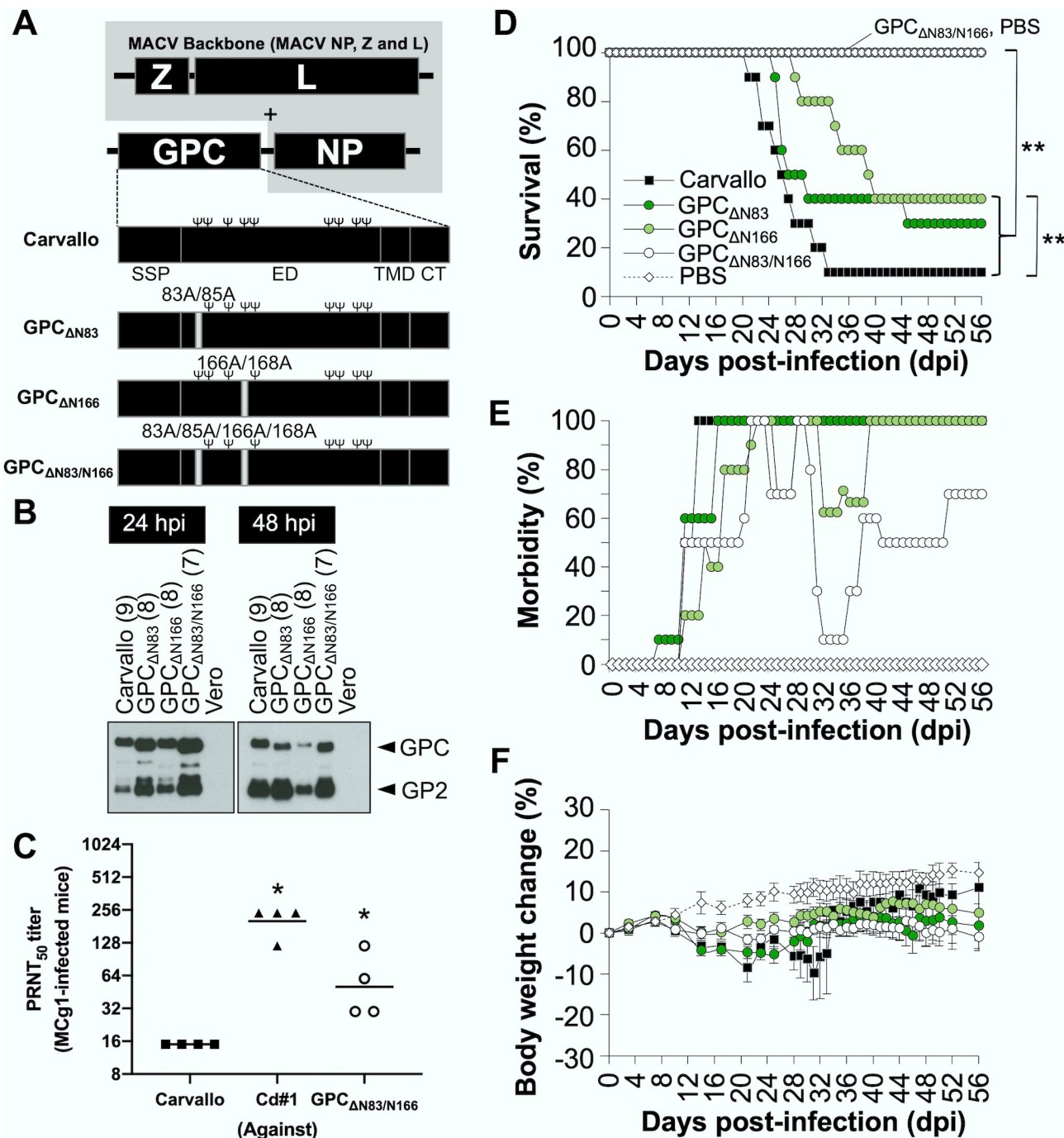

**Fig 4. Carvallo strains without the N-glycans on GPC.** (A) Schematic representation of the genome of MACVs. Ψ in black character represents an N-glycosylation site. (B) The representative WB for GPC. (C) Susceptibility of MACV $GPC_{\Delta N83/N166}$ to mouse antiserum to Cd#1. To determine the susceptibility of Carvallo, Cd#1 and MACV $GPC_{\Delta N83/N166}$ to the antiserum, the $PRNT_{50}$ was determined by neutralizing assay. Statistically significant differences compared with Carvallo are indicated by asterisks (*, $P < 0.05$ by Mann-Whitney U test). (D) Survival rate of the IFN-αβ/γ $R^{-/-}$ mice after infections (Carvallo N = 10, MACV $GPC_{\Delta N83}$ N = 10, MACV $GPC_{\Delta N166}$ N = 10, MACV $GPC_{\Delta N83/N166}$ N = 10, PBS N = 6). Statistically significant differences are indicated by asterisks (**, $P < 0.01$ by log rank test). (E) Morbidity rate of the IFN-αβ/γ $R^{-/-}$ mice after infections. (F) Body weight changes were monitored on the indicated days. Error bars indicate the SEM. The data shown are pooled from two independent experiments. The data of Carvallo- and PBS-inoculated mice in Fig 5D to 5F were overlapped with the data in Fig 1D–1F since these infectious experiments were performed together at the same time.

neutralization. In the next study, IFN-$\alpha\beta/\gamma$ R$^{-/-}$ mice were infected with MACV GPC$_{\Delta N83}$, MACV GPC$_{\Delta N166}$ and MACV GPC$_{\Delta N83/166}$ intraperitoneally and monitored for 56 days. The survival rates of MACV GPC$_{\Delta N83}$- and MACV GPC$_{\Delta N166}$-infected animals were moderately but significantly higher than that of wild-type Carvallo-infected animals (Fig 4D). Notably, all MACV GPC$_{\Delta N83/N166}$-infected mice survived and had the lowest morbidity (Fig 4D and 4E). Nevertheless, all of the mice developed mild disease manifestations such as scruffy coats and hunched posture between 21 dpi and 29 dpi, and most of them had progressive weight loss (Figs 4E, 4F and S1). This data suggests that the disruption of the N-glycosylation sites alone was not sufficient for full attenuation. Interestingly, the disease manifestations in MACV GPC$_{\Delta N166}$- and MACV GPC$_{\Delta N83/N166}$-infected mice were biphasic. The second phase occurred after 38 dpi and was characterized by neurological disease signs such as imbalance (Figs 4E and S1). These late neurological signs were similar to those of the surviving MCg1-infected mice.

We also studied virus dissemination in mice (Fig 5A). In MACV GPC$_{\Delta N83}$- and MACV GPC$_{\Delta N166}$ infections, only the animals that succumbed to infection or were euthanized prior to the study endpoint had detectable virus titers in their brains and lungs at a level markedly lower than these organs in Carvallo-infected animals. In MACV GPC$_{\Delta N83/N166}$-infected animals, we could not detect infectious virus in organs examined. However, we could detect viral RNA in the brains of all mice. No reversion to the wild-type sequence was observed for GPC$_{\Delta N83}$ and GPC$_{\Delta N166}$ in any sample.

We further performed PRNT$_{50}$ assays to determine if lack of N83 and N166 glycans on GPC increased MACV sensitivity to neutralizing antibodies. Sera from MACV GPC$_{\Delta N83}$, MACV GPC$_{\Delta N166}$, and MACV GPC$_{\Delta N83/N166}$-infected animals neutralized the homologous viruses more potently than they did Carvallo (Table 2), suggesting that GPC N-glycans at N83 and N166 both contributed to shielding neutralizing antibody epitopes of MACV GPC. Conversely, the neutralizing antibody activity against Carvallo was the highest in MACV GPC$_{\Delta N83/N166}$-infected animals, regardless of the collection day (Table 2 and Fig 3). Serum samples collected at 56 dpi from both MACV GPC$_{\Delta N166}$ and MACV GPC$_{\Delta N83/N166}$ groups had substantially higher neutralizing antibody activity against Carvallo than sera taken from the GPC$_{\Delta N83}$ group (Table 2 and Fig 3). Therefore, our data further suggests that the N-linked glycans at N166 and N83 impacted virus immunogenicity.

## MACV GPC$_{\Delta N83}$, MACV GPC$_{\Delta N166}$ and MACV GPC$_{\Delta N83/N166}$ have similar virus growth in cell culture

We characterized the growth kinetics of MACV GPC$_{\Delta N83}$, MACV GPC$_{\Delta N166}$ and MACV GPC$_{\Delta N83/N166}$ in Vero, C8D1A, and Neuro-2a cells (Fig 5B). Virus titers for MACV GPC$_{\Delta N83}$, MACV GPC$_{\Delta N166}$, and MACV GPC$_{\Delta N83/N166}$ at 96 hpi in Vero cells were slightly lower than that of Carvallo. Overall, we did not observe a substantial difference in virus growth due to lack of glycans at N83 or N166.

## Basis for glycan-mediated neutralization escape

We observed that glycans at N83 and N166 affected viral sensitivity to antibody neutralization. The arenavirus GPC is the sole viral protein present on virion surfaces and mediates receptor binding and virus entry. Neutralizing antibodies, in principle, can bind the GP1 receptor binding site (RBS) and prevent cellular attachment, or can bind to GPC and prevent conformational changes necessary for membrane fusion. To better understand how glycans at N83 and N166 could affect antibody neutralization, we used the coordinates of the trimeric LASV GPC to model MACV GPC (Fig 6). The glycan on MACV GPC N166 is proximal to the GP1 RBS

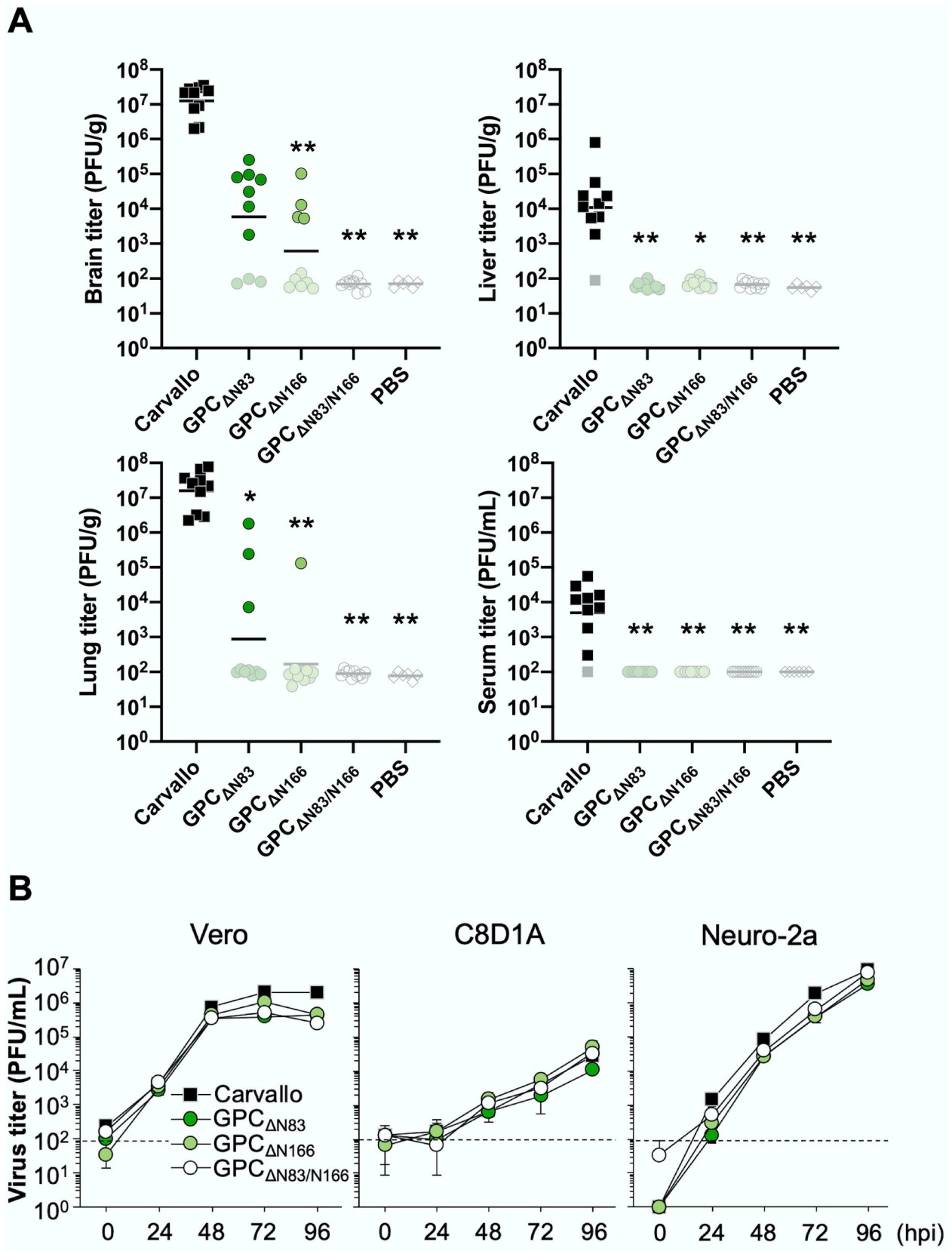

**Fig 5. Effects of the N-glycans on virus dissemination and growth curves.** (A) Virus titers in brain, lung, liver and serum. Samples under the detection limit were plotted with their detection limits in whitish color. The solid bar represents the geometric mean of the titers. The data of Carvallo- and PBS-inoculated mice in Fig 5A were overlapped with the data in Fig 2A since these infectious experiments were performed together at the same time. Statistically significant differences compared with Carvallo are indicated by asterisks (*, $P<0.05$ and **, $P<0.01$ by Dunn's post-test after Kruskal-Wallis test). (B) Virus growth of MACV $GPC_{\Delta N83}$, MACV $GPC_{\Delta N166}$ and MACV $GPC_{\Delta N83/N166}$ was characterized in Vero, C8D1A and Neuro-2a cells (MOI = 0.01). The dashed line indicates the detection limit. Data shown are averages of triplicate wells with error bars indicating the SD. The data of Carvallo in Fig 5B was overlapped with the data in Fig 2B since these infectious experiments were performed together at the same time.

for TfR1 [39] and could sterically hinder some antibodies from binding to the RBS ("site 1", Fig 6B). The glycan at position N83 is remote from the GP1 RBS but may sterically hinder antibody access to an epitope that spans both GP1 and GP2, as described for the LASV neutralizing antibody 37.7H ("site 2", Fig 6C). Although this neutralizing antibody epitope has not yet been described for New World arenaviruses, antibodies binding to this second site, like 37.7H, could prevent conformational changes required for membrane fusion.

## Discussion

In this study, we identified the pivotal roles of arenavirus GPC glycans in viral pathogenicity *in vivo*. Single substitutions F438I and F427I in GPC TMD have been identified as major attenuation factors in MACV and JUNV, respectively [31,34]. However, substitution F438I or F427I in GPC TMD alone was not able to completely attenuate the virulence of the viruses *in vivo* [31,34]. We previously reported that replacement of the ectodomain of MACV GPC with that of Cd#1-GPC (MCg1) also attenuated MACV, suggesting that the ectodomain of Cd#1 contains additional attenuation factors [37]. In this study, we found that the MCg1 chimeric virus frequently underwent mutations in animals to gain two glycans corresponding to the N83 and N166 glycosylation sites in MACV GPC. While MCg1 was partially attenuated in mice, acquisition of glycans at N83 and N166 rendered both rBI#6 and rBI#7 highly virulent in animals (Figs 1 and 2), to an extent similar to that of wild-type MACV. This could mean that the two glycans, especially the one at N166, were important for virulence. Conversely, loss of any one of the N-linked glycosylation sites (N83 or N166) led to the attenuation of the otherwise virulent MACV (Carvallo), a result further attesting to the critical roles of these glycans in pathogenicity (Figs 4 and 5). The survival rate and morbidity, which are important indicators for pathogenicity, were examined for correlation with the virus detection rate and production of neutralizing antibody (Fig 7). The data showed a statistically significant, negative correlation,

**Table 2. PRNT$_{50}$ geometric mean titer.**

| | Serum group | | | | |
|---|---|---|---|---|---|
| PRNT$_{50}$ geometric mean titer to | Carvallo | $GPC_{\Delta N83}$ | $GPC_{\Delta N166}$ | $GPC_{\Delta N83/N166}$ | PBS |
| Carvallo virus | 25.7[a] (N = 9, 29.3 dpi) [b] | 60 (N = 10, 37.3 dpi) | 111.1 (N = 9, 42.9 dpi) | 316.7 (N = 10, 56 dpi) | <30 (N = 6, 56 dpi) |
| Carvallo virus (by serum collected at 56 dpi) [c] | <30 (N = 1, 56 dpi) | 95.2 (N = 3, 56 dpi) | 285.4 (N = 4, 56 dpi) | 316.7 (N = 10, 56 dpi) | <30 (N = 6, 56 dpi) |
| $GPC_{\Delta N83}$ virus | NA[d] | >960 (N = 3, 56 dpi) | NA | NA | NA |
| $GPC_{\Delta N166}$ virus | NA | NA | >960 (N = 4, 56 dpi) | NA | NA |
| $GPC_{\Delta N83/N166}$ virus | NA | NA | NA | >960 (N = 10, 56 dpi) | NA |

a The titer <30 was calculated as 15 for geometric mean if samples contained <30. The titer was shown as <30 when all the samples were <30. The titer was shown as >XXX.X when samples contained >960.

b (Numbers of samples tested, the average sample collection days after infection).

c Only the data with serum collected at 56dpi was displayed to align the collection date.

d NA, Not available.

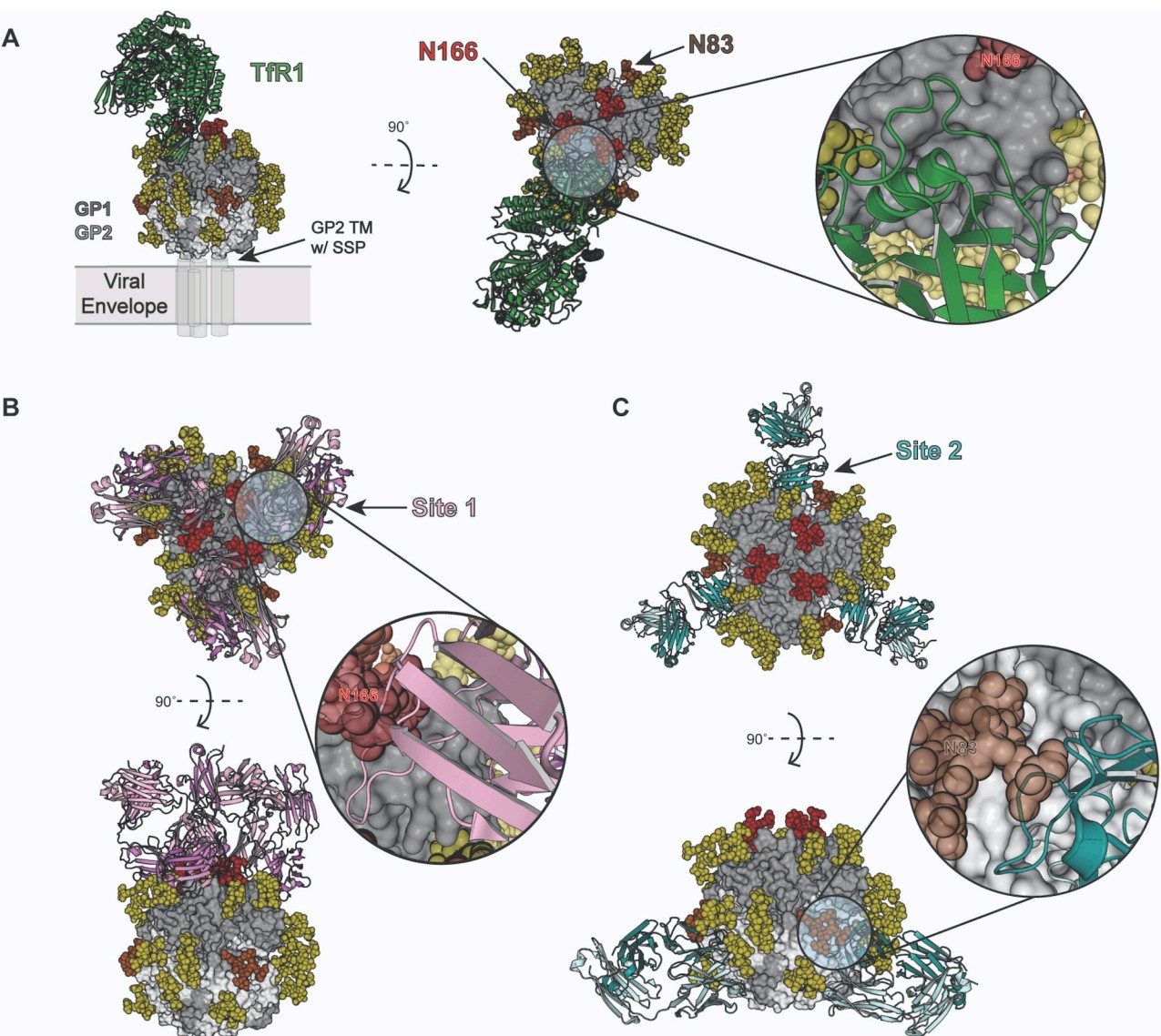

**Fig 6. N83/N166 glycan likely impairs antibody access to neutralizing epitopes on the arenavirus GPC.** (A) The ectodomain of MACV GPC was modelled by structural alignment of MACV GP1 (PDB ID: 3KAS) [39] to the LASV GPC trimer ectodomain (PDB ID: 5VK2) [58]] and is shown in top and side views. Surface renderings of the three GP1 protomers (dark gray) and the GP2 trimer (light gray) are shown. Carbohydrates were modelled as N-linked oligomannose-type structures and are shown as spheres. Glycans at position N83, N166, or at other positions are colored brown, red, and yellow, respectively. The TfR1 dimer (green) shown as a ribbon diagram bound to MACV GP1 was also derived from PDB ID: 3KAS. (B) As modelled, N166, which is near the GPC apex, would shield a neutralizing epitope ("site 1") that overlaps with the GP1 receptor-binding site. To model this, the MACV neutralizing CR1-07 Fab (PDB ID: 5W1M) [57] is shown in ribbon (pink) and docked onto the MACV GPC ectodomain. (C) An additional neutralizing epitope on the arenavirus GPC ("site 2"), which spans GP1/GP2 and adjacent protomers in LASV, could be affected by N83. The Fab for 37.7H (from PDB ID: 5VK2), a LASV neutralizing antibody, is shown in ribbon (teal) and docked onto the MACV GPC model ectodomain. Note that the modeling involves addition of glycan residues not visualized as clashing with CR1-07 or 37.7H Fab in PDB IDs 5W1M or 5VK2, respectively (in those structures, not all parts of the glycan had attributable electron density).

between the survival rate and the virus detection rate in the brain and lung. On the other hand, there was a statistically significant, positive correlation, between the morbidity in early infection (13 dpi) and late infection (37 dpi), and the virus detection rate in the brain and lungs. The survival rate and neutralizing antibody titer showed a significant positive correlation, while the neutralizing antibody titer and morbidity (37 dpi but not 13 dpi), showed a negative

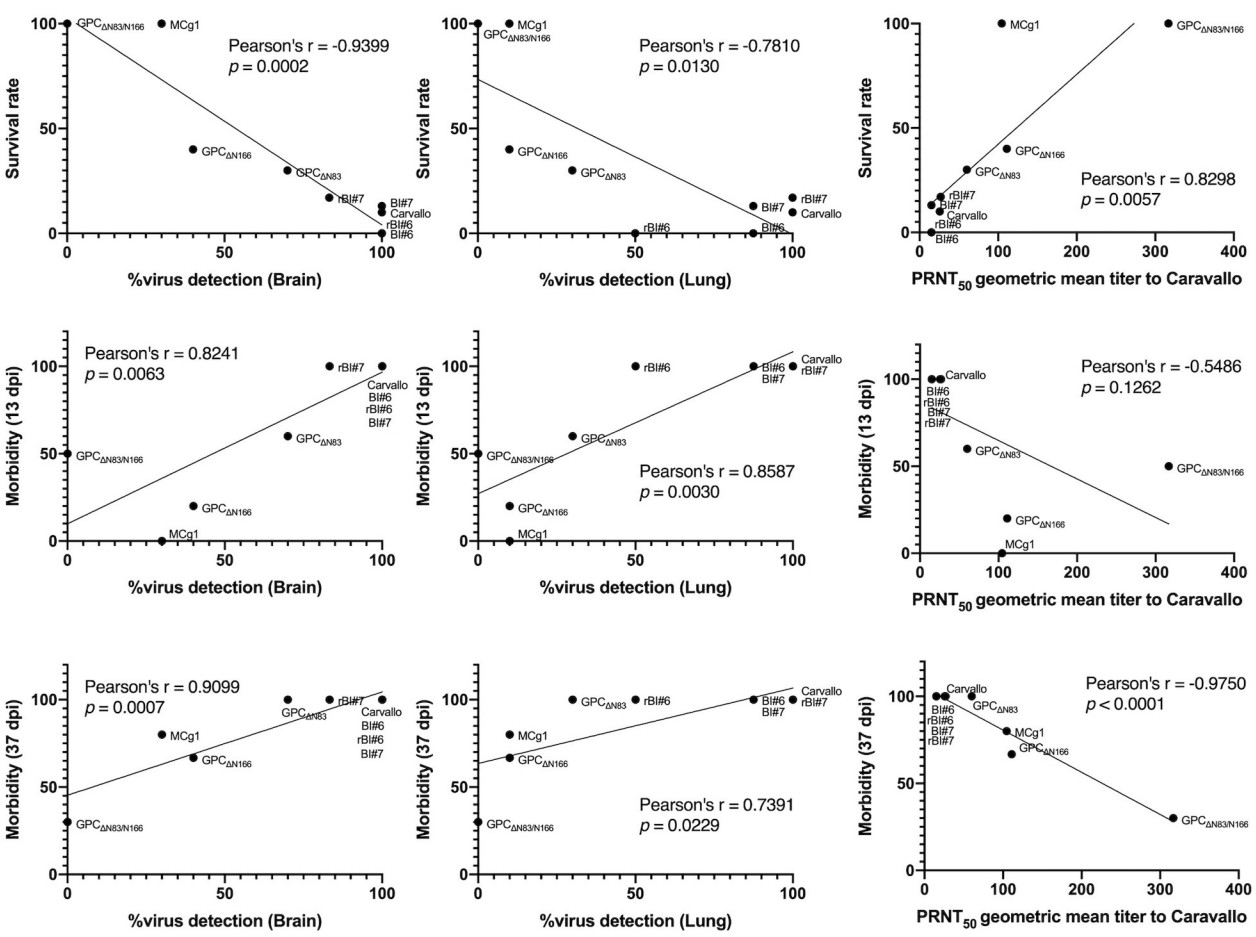

**Fig 7. Pearson correlation analysis for pathogenicity assessment.** Pearson correlation analysis shows a significant negative correlation between the survival rate and the virus detection rate in the highly susceptible organs (brain and lung), a significant positive correlation between the survival rate and the $PRNT_{50}$ titer, a significant positive correlation between the morbidity (13 dpi and 37 dpi) and the virus detection rate in brain and lung, and a significant negative correlation between the morbidity (37 dpi) and the $PRNT_{50}$ titer. Data at 13 dpi was selected as the early stage of infection since the morbidity of the MACV Carvallo-infected group reached 100% at 13 dpi. 37dpi was selected as the late stage of infection since there was a group with the survival rate of 0 at 38 dpi.

correlation. Therefore, the virus detection rate and neutralizing antibody titer are important factors for virus pathogenicity.

N-linked glycans at N83 and N166 were clearly critical for virus pathogenicity *in vivo*, while they had no apparent influence on virus growth in cultured cells. These results suggest a very strong selection pressure *in vivo*. In support of this, we found that mutant viruses acquiring GPC N-linked glycans at N83 and N166 emerged in a time-dependent manner in MCg1-infected IFN-αβ/γ $R^{-/-}$ mice. No substitutions were seen at 17 dpi. Later, 100% of the MCg1-infected mice contained a population of $GPC_{A168S/T}$ after 42 days of infection (S2 Table). Mutations were not observed after five serial passages of MCg1 *in vitro* [37] and did not affect MCg1 growth *in vitro*, which is important for development of a genetically stable and attenuated vaccine candidates [30]. No mutations occurred in MCg1 in cell culture experiments in our study. Nevertheless, a previous report showed that revertants appeared in serial passages of N-glycan deficient LCMV $GPC_{S116A}$, LCMV $GPC_{T234A}$ and LCMV $GPC_{S373A}$ mutants [24] in cell cultures. To minimize the likelihood of genetic reversion, we disrupted the glycosylation site by changing 4–5 nucleotides to mutate both N and S/T of the N-X-S/T motif, instead of

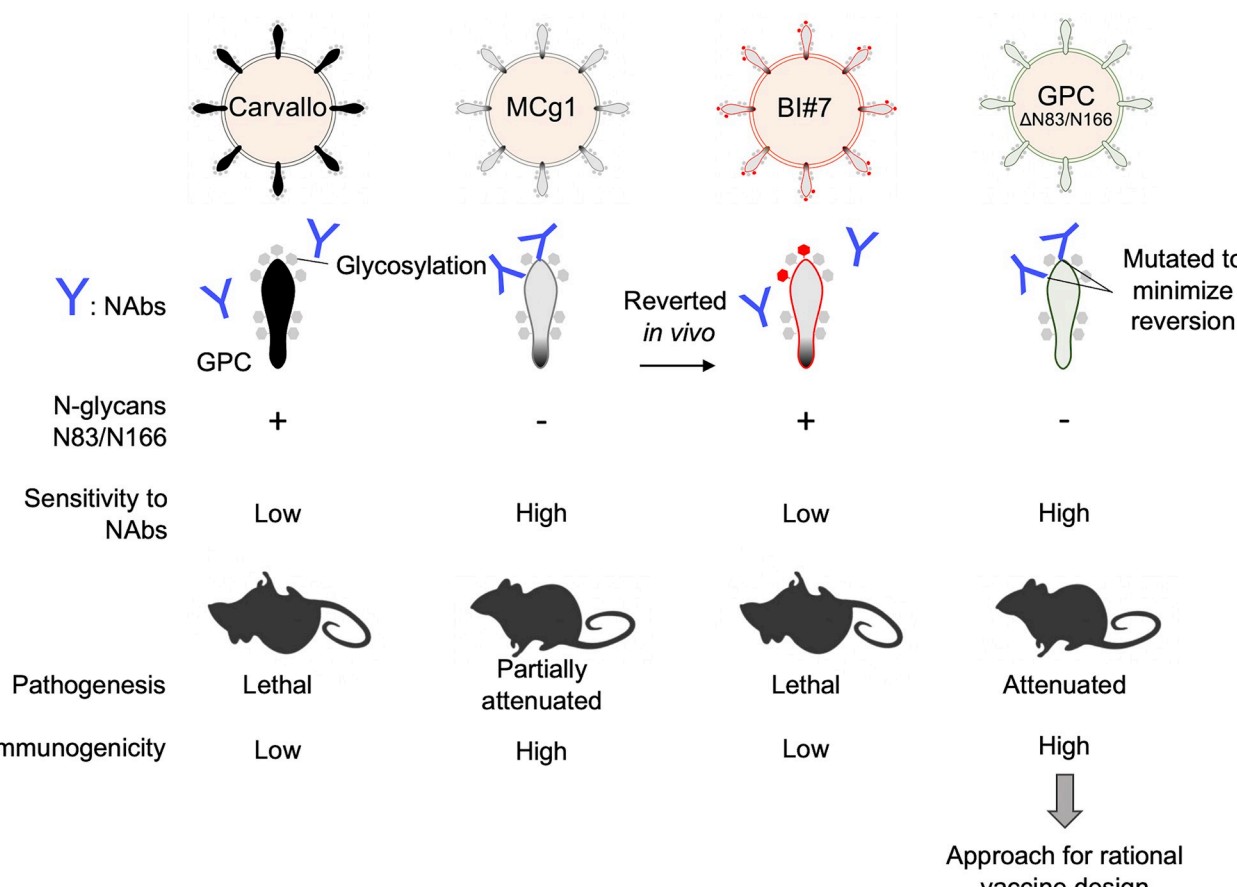

**Fig 8. The summary of the main findings of this study.** MCg1, which makes the GPC ectodomain of pathogenic MACV from Candid#1, is partially attenuated *in vivo*. MCg1 is highly sensitive to neutralizing antibodies, as is Candid#1. However, in the late stages of infection, the reversions to glycosylation at N83 and N166 occur highly frequently *in vivo*. When the revertant viruses were isolated and injected into mice, they showed high virulence and reduced susceptibility to neutralizing antibodies similar to wild-type MACV Carvallo. The pathogenicity of the virus MACV GPC$_{\Delta N83/N166}$, which has disrupted the glycosylation sequon at N83 and N166 to prevent reversion, is significantly reduced, indicating that the pathogenicity of the virus is significantly altered by the change in susceptibility to neutralizing antibodies. The infectious virus was undetectable in individuals, and its immunogenicity was higher than that of MACV Carvallo. Therefore, disrupting these specific critical glycosylations could be a reasonable approach for vaccine design.

disrupting one amino acid with a single nucleotide substitution (Fig 8). No revertant was observed in the animals infected with the viruses for as long as 56 days. This approach may be useful in designing an attenuated arenavirus vaccine.

An important observation from our study was the substantial impact of glycans at N83 and N166 on viral sensitivity to antibody neutralization, which is also supported by structural modelling of arenavirus receptor and antibody glycoprotein complexes (Fig 6). Viruses lacking both N83 and N166 glycans were most sensitive to neutralizing antibodies, indicating that removal of both glycans results in efficient virus neutralization (Figs 4 and 5) [25,26,40–43]. Comparing the frequencies of two reversions (GPC$_{P85S}$ and GPC$_{A168S/T}$) in MCg1, the reversion with A168S/T appeared more frequently than P85S (S1 and S2 Tables). Moreover, the N166 glycan had higher importance than N83 for lethality (low survival rate and high viral load) and low immunogenicity (Figs 4, 5 and 7 and Table 2). One possibility is that the N166 glycan is the predominant glycan covering the neutralizing epitope, while the N83 glycan is further away from it (Fig 6). Since MACV and JUNV do not use murine TfR1 as a receptor [44–47], further studies are need to clarify the contribution of GPC glycans in the context of

human TfR1. The crystal structure of MACV GP1 bound to human TfR1 [39] (Fig 6) support the idea that a higher importance of N166 glycan than that of N83 glycan is not limited to mice in MACV infection.

Additionally, our data for the first time indicated important roles of specific N-linked glycans in GPC immunogenicity. Infection with Carvallo lacking one or two N-linked glycans (MACV GPC$_{\Delta N83}$, MACV GPC$_{\Delta N166}$ and MACV GPC$_{\Delta N83/N166}$) induced higher neutralizing antibody titers against Carvallo than infection with Carvallo itself at 56 dpi (Table 2 and Fig 3). As shown in Fig 3B, the titers of neutralizing antibodies against Carvallo were significantly increased as a result of N166 deficiency (MACV GPC$_{\Delta N166}$). On the other hand, there is a trend that the N83 glycan deficient mutant (MACV GPC$_{\Delta N83}$) elicited higher levels of neutralizing antibody than MACV, although the difference was not significant. Further studies are warranted to investigate the contribution of these N-glycans on GPC immungenecity.

The difference in virulence of the viruses with or without the two glycans could be explained by differences in the viral resistance to neutralization and humoral immune responsiveness. In our previous study of MACV Carvallo infection in animals, the neutralizing antibody titers to MACV were below the detection limit at 4 weeks of infection, when the majority of infected mice died [29]. At the same time point, MCg1-infected animals developed neutralizing antibodies at titers of 1:42.4 (geometric mean, 33 dpi) against heterologous MACV Carvallo and 1:480 to >1:960 against homologous virus (see Table 2 of our previous publication [37]. These results support the hypothesis that the two N-glycans on GPC not only render the virus refractory to neutralizing antibodies but also negatively affect the host humoral immune response. In our recent study, we found that twenty non-neutralizing monoclonal antibodies against MACV GPC did not have an antibody-dependent cellular cytotoxicity (ADCC), and only neutralizing antibodies retained high ADCC reporter activity [9]. This indicates that neutralizing antibodies against MACV Carvallo control the virus not only by binding to the receptor binding sites but also by ADCC in infected individuals. Furthermore, non-neutralizing antibodies have been reported to contribute to reducing LCMV clearance mediated *via* a complement-dependent manner [48]. The removal of N-glycans may increase the sensitivity to antibodies *via* Fc-mediated effector functions, which should be investigated in future studies.

In a previous JUNV study, loss of glycosylation site at N166 did not seem to have an impact on virulence in intracranially infected suckling mice [31]. This discrepancy may be attributed to the difference in the maturity of humoral immunity and the route of infection [49,50]. In our experiments this glycan is clearly important for MACV dissemination and brain penetration. In the Albariño study this is certainly not the case as JUNV is directly inoculated into the brain so the virus doesn't require the ability to efficiently disseminate and penetrate the brain upon peripheral infection. Moreover, animals in our study have sufficient time to produce neutralizing antibodies before brain infection is established. Also, it is likely that once in the brain all of these viruses could grow to similar levels and cause disease but not all of them can reach the CNS if introduced peripherally [31].

Interestingly, the glycans on GPC seemed to affect MACV tropisim *in vivo*. In cultured cells, MACVs with or without specific N-glycosylation sites on GPC grew similarly in kidney- or brain-derived cell lines (Figs 2 and 5). However, there was clear differences in viral loads in the organs of infected animals (Figs 2A and 5A). BI#7/rBI#7, which had the same number of N-glycans as Carvallo, showed the highest viral load among the mutants in brain and lung, which was only slightly lower than that of wild-type Carvallo. This may be because of the difference in the position of the third glycan site between these viruses. The GPC ectodomain of BI#7/rBI#7 is in the Cd#1 backbone, and the third glycan of BI#7/rBI#7 GP1 is located at N105. On the other hand, MACV Carvallo GP1 has the third glycan at N137. Overall, the loss of glycans at N83 and/or N166 resulted in lower viral loads in organs tested in our study. N-

glycans on GPC have been found to affect the tissue tropism of Old World LCMV [25]. Further studies are required to understand how N-glycans on GPC alter the tissue tropism of New World arenaviruses.

Recently we have demonstrated that JUNV GPC lacking the N-glycosylation site at N166 accumulates in the form of dimers and trimers and is more readily degraded than the wild-type GPC within the lysosome [32,33]. This phenomenon likely leads to increased antigen presentation and pro-inflammatory responses [32,33]. Based on this finding and the importance of the N-glycans of GPCs in proper folding and cleavage of GPCs [20–22,25], wild-type MACV GPCs with glycans at N83/N166 are likely being folded properly and therefore are less likely be degraded and induce an immune response. In addition, wild-type MACV GPC could be more efficiently incorporated into the particles, resulting in an increase in the density of the GP1/GP2 complex in the virions. A higher density of envelope glycoprotein in Ebola virus virions enhances the lectin receptors' attachment and also enables a higher density of GP to escape from neutralizing antibodies [51]. Thus, the density of the GP1/GP2 complex in virions should be investigated in the future to understand if it is influenced by N-glycosylations.

In MCg1-infected mice, although mutants with the glycosylation(s) at N83 and N166 emerged in the majority of animals, those mice did not succumb to infection like those infected by rBI#6 and rBI#7. It is likely that neutralizing antibodies elicited by MCg1 facilitated hosts to control mutant virus infections as a result of the race between neutralizing antibody production and the emergence of mutant viruses. In closely related Tacaribe virus infection, neutralizing antibodies was found to be produced at 8 dpi in IFN-$\alpha\beta/\gamma$ R$^{-/-}$ mice [52]. In our study, mutants were not detected at 17 dpi. It is very likely that neutralizing antibodies were elicited before the emergence of mutant viruses in MCg1-infected animals. Although those mutants acquired additional glycans that rendered viruses less susceptible to neutralizing antibodies, data in Fig 3 showed MCg1-infected mice developed antibody responses that could neutralize the mutants (rBI#6 and rBI#7) to some extent (>1:128). Thus, it is not unexpected that hosts survived despite the appearance of those mutants that otherwise are more virulent. In some animals, especially those infected by MCg1, MACV GPC$_{\Delta N166}$ and MACV GPC$_{\Delta N83/N166}$, the diseases appeared to be alleviated or not overt temporarily and then exacerbated or appeared later (S1 Fig). The mechanism remains to be investigated in future studies. One possibility is that infections were transiently controlled by antibodies and then worsened probably due to virus dissemination to the immune-privileged brain.

In conclusion, removal of specific N-linked glycans from glycoprotein may attenuate pathogenic arenaviruses by increasing immunogenicity and viral sensitivity to neutralizing antibodies due to the loss of a glycan shield. Additionally, those N-glycans may also affect viral tropism *in vivo*. It is worth investigating if including the GPC TMD F438I mutation and the recently reported IGR deletion mutation in MACV L segment [53] will further attenuate MACV. Taken together, our findings may provide insight into vaccine design for highly pathogenic arenaviruses.

## Materials and methods

### Ethics statement

All experiments with live viruses except Cd#1 were performed in the BSL-4 laboratories at the Galveston National Laboratory (GNL) in accordance with institutional safety guidelines, NIH guidelines and US federal law. Animals were humanely euthanized at the end of study (56 dpi) or if they became paralyzed or lost more than 20% of the body weight. The Institutional Animal Care and Use Committee at the University of Texas Medical Branch at Galveston approved the study protocol (1208050A).

## Cells and viruses

Vero cells (ATCC, CCL-81), Neuro-2a (ATCC, CCL-131), C8D1A (ATCC, CRL-2541) and baby hamster kidney (BHK-21) cells (ATCC, CCL-10) were maintained in minimal essential medium (MEM) (GE Healthcare Life Sciences) supplemented with 10% FBS (Life Technologies) and 1% Penicillin-Streptomycin (P/S)(Life Technologies) at 37˚C with 5% $CO_2$. HEK-293T cells (Life Technologies) were maintained in Dulbecco's modified eagles medium (DMEM) (GE Healthcare Life Sciences) with 10% FBS (Life Technologies) and 1% P/S (Life Technologies) at 37˚C with 5% $CO_2$.

The recombinant MACV Carvallo strain (Carvallo) [54], recombinant Cd#1 (Cd#1) [55], MCg1 [37], rBI#6, rBI#7, MACV $GPC_{\Delta N83}$, MACV $GPC_{\Delta N166}$, MACV $GPC_{\Delta N83/N166}$ were rescued by using a reverse genetics system previously described [55,56]. We used the first passage of viruses for all infection experiments in this study, except isolates (BI#6 and BI#7). *In vitro* infection, confluent monolayers of Vero, C8D1A or Neuro-2a cells were infected with viruses at MOI = 0.01 for virus growth curves or MOI = 1 for protein expression profiling. After 1h incubation at 37˚C with 5% $CO_2$, medium was replaced with MEM containing 2% FBS and 1% P/S. The supernatants or cells were collected at the indicated days. Virus titers of serum samples, homogenized organs or supernatant from cell culture were determined by plaque assay with Tragacanth Gum as previously described [37].

## Animal studies

IFN-αβ/γ R-/- mice were bred and maintained in the ABSL-2 facilities in the GNL at the University of Texas Medical Branch at Galveston. To see the effect of the GPC glycosylations at N83 and N166 on virulence, 5 to 9 week-old IFN-αβ/γ R-/- mice were challenged by intraperitoneal injection with brain isolates and other recombinant MACVs (10,000 PFU) and monitored for 56 days post-infection (dpi). The animal experiments were performed twice in independent studies except for rBI#6 and rBI#7. Serum, brain and liver samples were collected for virus titration and virus RNA when animals were euthanized or dead. To obtain anti-Cd#1 serum samples, 7 to 8 week-old C57BL/6 (Jackson Laboratory) mice were immunized *via* intraperitoneal injection with Cd#1 (10,000 PFU). Serum samples were collected at 35 dpi.

## RNA extraction, cDNA synthesis and sequence analysis

RNAs were extracted from homogenized organs or lysed cell lines by using Trizol reagent (Life Technologies) and Direct-zol RNA MiniPrep kits (Zymo Research, Irvine, CA) as previously described [36]. Reverse transcription was performed by using the Superscript III First-Strand Synthesis System (Life Technologies) with random primers according to the manufacturer's protocol. cDNAs for GPC ORF were amplified by PCR with 5'-CGC ACCGGGGATCCTAGGCGATTC-3' and 5'-CCTCTCAGCCTTCTATTTCTACCC-3'. cDNAs for whole virus genome were amplified by PCR into two and three DNA fragments for viral S with previously described primers [37] and L segments with the following primers, respectively: 5'- CGCACCGGGGATCCTAGGCGTAAC-3' and 5'-TAGGAACTGTGCCA GAAAGG-3', 5'-TCTCTAACGCACTTGCTACC-3' and 5'-TTGGATGTGCTGTGGTGA AC-3', 5'-GAGGATGTTGCCTAACTC-3' and 5'-CGCACCGAGGATCCTAGGCGACAC-3'. To read the sequence of the 5' and 3' end, the 5' RACE System for Rapid Amplification of cDNA Ends (Life Technologies) and the 3' RACE System for Rapid Amplification of cDNA Ends (Life Technologies) were used according to the manufacturer's protocol. After purification using a QIAquick PCR purification kit (Qiagen), PCR products were directly sequenced using an ABI Prism 3130xl DNA sequencer (Life Technologies).

## TA cloning

To read the sequence of the N-glycosylation sites at N83 and N166 on GPC of MCg1 recovered from infected mice, the ORF of GPC amplified with 5'-CGCACCGGGGATCCTAGGCGAT TC-3' and 5'-CCTCTCAGCCTTCTATTTCTACCC-3' was cloned into the pCR2.1-TOPO TA vector (Life Technologies) according to the manufacturer's protocol. The purified plasmids were directly sequenced as mentioned above.

## Virus isolation

BI#6 and BI#7 were isolated from brains of MCg1-infected mice. The brains were homogenized in MEM containing 2% FBS and 1% P/S. After amplification in Vero cells, the viruses were serially diluted and inoculated onto Vero cell monolayers in 12-well plates. After incubation for 1h at 37˚C with 5% $CO_2$, the inoculum was replaced by 0.5% agarose overlay medium and incubated for 8 days at 37˚C with 5% $CO_2$. Plaques were stained with 0.33% Neutral red in PBS and incubated for 2h. The plaques were picked up and resuspended in 500 μl of MEM containing 2% FBS and 1% P/S. Then, the virus was amplified in Vero cells and the plaque purification was repeated. The resuspended virus was amplified in Vero cells and the supernatant was used as working stock after whole genome sequencing, as described above.

## Construction of S segments for rBI#6 and rBI#7

BI#6 and BI#7 have synonymous substitutions other than in the triplet sequence for the N-glycosylation in the S segment as well as in the L segment. To exclude the synonymous substitutions from the S segments of BI#6 and BI#7, the fragments were amplified using Platinum Pfx DNA polymerase (Life Technologies) with the following primers: 5'-CGCACCGGGGATCC TAGGCGATTC-3' and 5'-CCTGCCTGTCCGAATACTCTTTGC-3'. After purification, the PCR products and the pRF42-MCg1 S seg were digested with EcoRI-HF and HindIII. The synonymous substitutions were excluded from the fragments since the synonymous substitutions were located outside of the restriction enzyme sites. The digested PCR products and the vector were removed from agarose gel and purified. Then, the PCR products were ligated with the vector. The ligation products were transformed into DH5α competent cells and then plated on LB plates containing 100 μg/mL of ampicillin. After screening for the correct sequence, the plasmids were amplified in DH5α competent cells and purified with EndoFree Plasmid Maxi Kit (QIAGEN). To make rBI#6 and rBI#7 without the silent mutation in the L segment, pRF42-MACV L seg, which has the correct sequence, was used upon the recovery using the reverse genetics system.

## Western blotting

Western blotting was performed as previously described [32]. Briefly, infected cells were harvested in the 2x Laemmli Sample Buffer (Bio-Rad) containing 5% β-mercaptoethanol and boiled at 95˚C for 5 min. The samples were loaded in the wells of 4–15% Mini-Protean TGX gels (Bio-Rad). Then, proteins were transferred to polyvinyl difluoride (PVDF) membranes using the Trans-Blot Turbo Transfer System (Bio-Rad). After blocking with PBS containing 0.05% Tween 20 (PBS-T) and 5% skim milk (blocking buffer) for 1h at room temperature, the membranes were incubated with primary antibodies in blocking buffer at 4˚C overnight. After three washes with PBS-T, the membranes were incubated with secondary antibody in blocking buffer for 2 h at room temperature. After three washes with PBS-T, the HRP signal was visualized by enhanced chemiluminescence (ECL) (ECL Western Blotting System, Amersham). The

primary polyclonal antibodies targeting the cytoplasmic tail of MACV GP2 were created by immunization with synthetic peptides (KYPRLKKPTIWHKR) (ProSci) in rabbits.

## Plaque reduction neutralization test

Serum samples were heat inactivated at 56˚C for 30 min. The serum samples were diluted (final dilution range, 1:30 to 1:960) and mixed with an equal volume of diluent containing 80 PFU of each virus. After incubation for 1 h at 37˚C, the mixture was applied on Vero cell monolayers. After the incubation for 1 h at 37˚C, the inoculum was replaced with tragacanth overlay (1.2% tragacanth gum mixed with equal volume of Temin's 2X MEM containing 4% FBS and 2% P/S) and incubated for 7 to 8 days. Then the plates were fixed and stained with 1% crystal violet in 10% formalin.

## Statistical analysis

Data were analyzed using Dunn's post-test after Kruskal-Wallis test, log rank analysis, Pearson correlation analysis and Mann-Whitney U test. Results were considered to be statistically significantly different when the P value was <0.05.

## Molecular modeling of MACV GPC

The ectodomain of MACV GPC was modelled using PyMol Molecular Graphics System, Schrödinger, LLC, by structural alignment of MACV GP1 (PDB ID: 3KAS) [57] to the LASV GPC trimer (PDB ID: 5VK2) [58] coordinates. Three copies of MACV GP1 were grafted onto the LASV GPC trimer by aligning the respective GP1 subunits. LASV GP1 coordinates that overlapped with MACV GP1 were then removed, and GP1 residues lacking in MACV GP1 were modelled as observed in 5VK2. The final model includes GP1 residues T59-P83 and P242-S259 from 5VK2, GP1 residues N86-H241 from 3KAS, and the LASV GP2 coordinates from 5VK2. Glycans were modelled as oligomannose-type structures using Coot[1][59]. Modelling of the human TfR1 ectodomain bound to MACV GPC was performed by aligning MACV GP1/TfR1 coordinates (PDB ID: 3KAS) onto the MACV GPC model. Modelling of the 37.7H Fab bound to MACV GPC was performed by alignment of LASV GPC-37.7H Fab coordinates (PDB ID: 5VK2) onto the MACV GPC model. Modelling of the CR1-07 Fab bound to the MACV GPC model was performed by alignment of the MACV GP1/CR1-07 Fab coordinates (PDB ID 5W1M) onto the MACV GPC model [57].

## Supporting information

**S1 Fig. Percentage of symptoms in infected animals.** The heat map color and the indicated numbers represent the percentage of each clinical sign in animal groups.
(TIF)

**S2 Fig. Neutralizing activity of sera from MCg1-infected mice against parental and mutant MCg1.** This graph represents the percentage of plaque numbers for each serum dilution.
(TIF)

**S1 Table. Frequency of amino acid sequence substitutions in GPC of MCg1 in animals.** [a] No mutation was observed at the position.
(DOCX)

**S2 Table. Summary of amino acid changes in GPC of MCg1 in this study.** [a]"Yes" indicates the population contains the substitution at the position. [b]"No" indicates the population has a

racesingle peak in the original sequence chromatograms.
(DOCX)

## Author Contributions

**Conceptualization:** Takaaki Koma, Cheng Huang, Slobodan Paessler.

**Funding acquisition:** Slobodan Paessler.

**Investigation:** Takaaki Koma, Adrian Coscia, Steven Hallam, John T. Manning, Junki Maruyama, Aida G. Walker, Milagros Miller, Jeanon N. Smith.

**Methodology:** Takaaki Koma, Cheng Huang, Slobodan Paessler.

**Resources:** Michael Patterson.

**Software:** Adrian Coscia, Jonathan Abraham.

**Supervision:** Slobodan Paessler.

**Validation:** Takaaki Koma, Junki Maruyama.

**Writing – original draft:** Takaaki Koma, Cheng Huang.

**Writing – review & editing:** Takaaki Koma, Cheng Huang, Junki Maruyama, Jonathan Abraham, Slobodan Paessler.

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
