## [Decision Letter · Decision Letter 0]

21 Nov 2020

Dear Dr. Paessler,

Thank you very much for submitting your manuscript "Glycoprotein N-linked glycans play a critical role in arenavirus pathogenesis" for consideration at PLOS Pathogens. As with all papers reviewed by the journal, your manuscript was reviewed by members of the editorial board and by several independent reviewers. In light of the reviews (below this email), we would like to invite the resubmission of a significantly-revised version that takes into account the reviewers' comments.

I also read the manuscript, and have several additional comments:

1. The manuscript needs editing for English usage and grammar. Articles are missing or appear incorrectly in many instances, tenses are incorrect, as well as other issues (e.g. "reversed the T168A substitution on Cd#1 GP1" should be in).

2. p. 8, line 131. Fig. 1B does not show that the nucleotide substitutions were found in almost all infected animals. In fact, if I understand correctly, 3/4 animals maintained the MCg1 sequence. 4/7 is hardly all. I also didn't understand why those animals that had mutant viruses did not fall ill if these viruses were more pathogenic and resistant to neutralizing antibodies.

4. I found the morbidity curves a bit strange, because they go up and down (1E and 3E). Does this mean that some animals that showed signs of morbidity improved, and then later succumbed to disease.

We cannot make any decision about publication until we have seen the revised manuscript and your response to the reviewers' comments. Your revised manuscript is also likely to be sent to reviewers for further evaluation.

Sincerely,

Susan R. Ross, PhD

Section Editor

PLOS Pathogens

Susan Ross

Section Editor

PLOS Pathogens

Kasturi Haldar

Editor-in-Chief

PLOS Pathogens

orcid.org/0000-0001-5065-158X

Michael Malim

Editor-in-Chief

PLOS Pathogens

orcid.org/0000-0002-7699-2064

Reviewer's Responses to Questions

**Part I - Summary**

Reviewer #1: Koma et al. report in this manuscript that N-lycans at MACV GPC 83 and 166 contribute to virus virulence by evading humoral immunity. The authors first observed that a chimeric MACV containing a Candid #1 GPC ectodoman lacking N-glycans at those two sites restored those glycans when they replicated in mice. They introduced these glycosylations into the parental strain and observed much higher mortality and tissue viral loads. When these glycosylation sites were mutated in MACV GPC preventing reversion, the virus was significantly attenuated in mice. The authors went ahead and showed the enhanced virulence of the gain-of-glycan mutants at least partially derived from the decreased sensitivity to neutralizing sera. This study in general is well conducted and follows logical sequence. Some suggestions, intended to clarify the interpretation of the study, are given below.

Reviewer #2: Koma T. et al. presents an analysis of two N-linked glycans and their role in the attenuation and / or virulence of Candid#1 and MACV. The starting point for this study is revertants that were isolated from animals infected with a chimeric MACV that had the Candid#1 ectodomain. The reversions restored two N-linked glycans at positions N83 and N166 and made the isolates more virulent and less susceptible to neutralizing antibodies. Elimination of these specific glycans from the MACV spike complex attenuated the virus. Also, the authors show that sera from animals infected with MACV that his spike is lacking the two glycans neutralize MACV-Carvallo strain more efficient compared to sera from animals that were infected by either Carvallo or by a mutant that is lacking only one of the glycans.

Overall, this is a well performed study with significant new findings that may contribute to the development of attenuated New World Arenaviral strains that could be used for vaccination.

By addressing the comments below the manuscript could be further improved.

Reviewer #3: The author have revisited an old finding in the biology of the arenaviruses. It has been known since the early days of studies of the cell biology of the group that glycosylation plays an important, even crucial role in the biosynthesis and expression of cogently folded GPC. From the work of Wright in the late 1980's who showed that glycosylattion of the GPC precursor was requires to express properly folded and immunoreactive GPC and to cleave the precursor to form the mature GP1/GP2 complex and for neutralizing antibody recognition. This was followed by the work of Bonhomme et al who showed in two papers published in 2011 and 2013 that glycosylation influenced both viral fitness and Tropism, and GP expression and function. Later work by Sommerstein and published in PLoS Pathogen demonstrated that the evasion from neutralizing antibody and virus persistence was promoted by a "glycan Sheld", and in a second paper, Watanabe et al showed that the Lassa Glycan Shield provides a model for Immunologic resistance.

So the impact of the findings herein are quite predictable, and I have to ask whet is really new. It appears to this reviewer that this novel infornmation ois limited to the the non -conceptual information that is conveyed in the results, and certainly not to the concept that viral glycoprotein glycosylation is capable of an immunomodulating activity.

**Part II – Major Issues: Key Experiments Required for Acceptance**

Reviewer #1: 1.The authors associated increased virulence of the N83/N166 virus with its increased resistance to neutralization. Please discuss if this means the differences in the virulence between rBI#7 (or MACV) and MCg1 (or Candid#1) may not be revealed until humoral immunity is generated. Humans die in ~14 days after the onset of symptoms.

2. The authors mentioned that the mutations restoring N166 glycan were more frequent than those restoring N83 glycan (lines 372-373) and that N166 glycan is more important for shielding neutralizing antibody epitopes (i.e. of the receptor-binding site) than the glycan at N83 (lines 273-274 and 294-295). Interestingly, while this phenomenon was observed in mice, mTfR1 is not believed an efficient receptor for MACV (PMC2268193). The authors can add this point in Discussion.

3. The authors also suggested increased proinflammatory cytokines, induced by increased degradation of the GPC, as a mechanism for lower virulence of the viruses lacking N166. Could higher abundance of GPC on the virion explain higher dissemination and virulence of the viruses containing N83/N166? This point should also be discussed.

Reviewer #2: I have only one major comment:

Although neutralization is clearly an important function of antibodies, Fc-mediated effector functions may also play an important role in suppressing viremia. The removal of glycans can render the virus more sensitive to neutralization as the authors propose but it may also facilitate binding of antibodies in geometries that can better recruit effector functions. The authors could address this possibility by performing PRNT in the presence or absence of complement, for example. In addition, it will be worth discussing this possibility in the discussion.

Reviewer #3: The paper is written with no historical context. With proper framing the findings of specific mutations and their effects on pathogenesis could be of interest. But to achieve that aim the manuscript needs a total re-write with the incorporation of a consideration of the known universe of findings. The potential novelty that the group could offer is the analysis of the high-containment pathogenesis of the various mutants. But this needs to be done according to state of the art standards with the incorporation of parameters of molecular pathogenesis and histopathology to differentiate the study from another run-of-the-mill pathology manuscript.

**Part III – Minor Issues: Editorial and Data Presentation Modifications**

Reviewer #1: Changing “Group” to “Plasma group” or “Serum group” will help to understand Tables 1 and 2. “Carvallo, 56 dpi” also needs an explanation.

Reviewer #2: Minor issues:

1) The two PRNT tables are not sufficiently clear since the sera sources and the target viruses have the same names. Please clearly state the source of sera and the target virus in the PRNT.

2) I couldn’t find a reference to Figure 7 in the text.

3) At some places the wrong references are being called (line 380 for example), please double check.

4) I suggest to move figure S2B to the main text, as this figure shows some of the more significant results of this study.

Reviewer #3: (No Response)

PLOS authors have the option to publish the peer review history of their article (what does this mean?). If published, this will include your full peer review and any attached files.

Reviewer #1: No

Reviewer #2: No

Reviewer #3: No
---

## [Editor Report · Decision Letter 1]

29 Jan 2021

Dear Dr. Paessler,

Thank you very much for submitting your manuscript "Glycoprotein N-linked glycans play a critical role in arenavirus pathogenicity" for consideration at PLOS Pathogens. I read the revision and am happy with the changes to the manuscript. There is one small additional change that I think is needed. You responded to one of the original comments regarding the use of mouse TfR1 as a receptor "**Radoshitzky reported that MACV does not use murine TfR1 as a receptor as efficiently as human TfR1 in their experiments using pseudotype virus (****44****)."**

I think it's pretty clear that MACV (and JUNV) does not use mouse TfR1 as a receptor, not just that it is inefficient; this has been shown with replication-competent viruses as well as pseudoviruses (please cite references PMCID: PMC7297529 and PMCID: PMC3233171). I would rephrase this section to say "MACV doesn't not use murine TfR1 as a receptor. Further studies are need to clarify the contribution of GPC glycans...."

Once you modify the manuscript accordingly, we would be happy to accept it.

Sincerely,

Susan R. Ross, PhD

Section Editor

PLOS Pathogens

Susan Ross

Section Editor

PLOS Pathogens

Kasturi Haldar

Editor-in-Chief

PLOS Pathogens

orcid.org/0000-0001-5065-158X

Michael Malim

Editor-in-Chief

PLOS Pathogens

orcid.org/0000-0002-7699-2064
---

## [Editor Report · Decision Letter 2]

3 Feb 2021

Dear Dr. Paessler,

We are pleased to inform you that your manuscript 'Glycoprotein N-linked glycans play a critical role in arenavirus pathogenicity' has been provisionally accepted for publication in PLOS Pathogens.

Best regards,

Susan R. Ross, PhD

Section Editor

PLOS Pathogens

Susan Ross

Section Editor

PLOS Pathogens

Kasturi Haldar

Editor-in-Chief

PLOS Pathogens

orcid.org/0000-0001-5065-158X

Michael Malim

Editor-in-Chief

PLOS Pathogens

orcid.org/0000-0002-7699-2064
---

## [Editor Report · Acceptance letter]

24 Feb 2021

Dear Dr. Paessler,

We are delighted to inform you that your manuscript, "Glycoprotein N-linked glycans play a critical role in arenavirus pathogenicity," has been formally accepted for publication in PLOS Pathogens.

Best regards,

Kasturi Haldar

Editor-in-Chief

PLOS Pathogens

orcid.org/0000-0001-5065-158X

Michael Malim

Editor-in-Chief

PLOS Pathogens

orcid.org/0000-0002-7699-2064